# Combinatorial Optimization with Graph Convolutional Networks and Guided Tree Search

**Zhuwen Li**
Intel Labs

**Qifeng Chen**
HKUST

**Vladlen Koltun**
Intel Labs

## Abstract

We present a learning-based approach to computing solutions for certain NP-hard problems. Our approach combines deep learning techniques with useful algorithmic elements from classic heuristics. The central component is a graph convolutional network that is trained to estimate the likelihood, for each vertex in a graph, of whether this vertex is part of the optimal solution. The network is designed and trained to synthesize a diverse set of solutions, which enables rapid exploration of the solution space via tree search. The presented approach is evaluated on four canonical NP-hard problems and five datasets, which include benchmark satisfiability problems and real social network graphs with up to a hundred thousand nodes. Experimental results demonstrate that the presented approach substantially outperforms recent deep learning work, and performs on par with highly optimized state-of-the-art heuristic solvers for some NP-hard problems. Experiments indicate that our approach generalizes across datasets, and scales to graphs that are orders of magnitude larger than those used during training.

## 1 Introduction

Many of the most important algorithmic problems in computer science are NP-hard. But their worst-case complexity does not diminish their practical role in computing. NP-hard problems arise as a matter of course in computational social science, operations research, electrical engineering, and bioinformatics, and must be solved as well as possible, their worst-case complexity notwithstanding. This motivates vigorous research into the design of approximation algorithms and heuristic solvers. Approximation algorithms provide theoretical guarantees, but their scalability may be limited and algorithms with satisfactory bounds may not exist [3, 38]. In practice, NP-hard problems are often solved using heuristics that are evaluated in terms of their empirical performance on problems of various sizes and difficulty levels [15].

Recent progress in deep learning has stimulated increased interest in learning algorithms for NP-hard problems. Convolutional networks and reinforcement learning have been applied with inspiring results to the game Go, which is theoretically intractable [34, 35]. Recent work has also considered classic NP-hard problems, such as Satisfiability, Travelling Salesman, Knapsack, Minimum Vertex Cover, and Maximum Cut [37, 6, 10, 32, 25]. The appeal of learning-based approaches is that they may discover useful patterns in the data that may be hard to specify by hand, such as graph motifs that can indicate a set of vertices that belong to an optimal solution.

In this paper, we present a new approach to solving NP-hard problems that can be expressed in terms of graphs. Our approach combines deep learning techniques with useful algorithmic elements from classic heuristics. The central component is a graph convolutional network (GCN) [12, 24] that is trained to predict the likelihood, for each vertex, of whether this vertex is part of the optimal solution. A naive implementation of this idea does not yield good results, because there may be many optimal solutions, and each vertex could participate in some of them. A network trained without provisions that address this can generate a diffuse and uninformative likelihood map. To overcome this problem, we use a network structure and loss that allows the network to synthesize a diverse set

of solutions, which enables the network to explicitly disambiguate different modes in the solution space. This trained GCN is used to guide a parallelized tree search procedure that rapidly generates a large number of candidate solutions, one of which is chosen after subsequent refinement.

We apply the presented approach to four canonical NP-hard problems: Satisfiability (SAT), Maximal Independent Set (MIS), Minimum Vertex Cover (MVC), and Maximal Clique (MC). The approach is evaluated on two SAT benchmarks, an MC benchmark, real-world citation network graphs, and social network graphs with up to one hundred thousand nodes from the Stanford Large Network Dataset Collection. The experiments indicate that our approach substantially outperforms recent state-of-the-art (SOTA) deep learning work. For example, on the SATLIB benchmark, our approach solves all of the problems in the test set, while a recent method based on reinforcement learning does not solve any. The experiments also indicate that our approach performs on par with or better than highly-optimized contemporary solvers based on traditional heuristic methods. Furthermore, the experiments indicate that the presented approach generalizes across datasets and scales to graphs that are orders of magnitude larger than those used during training.

## 2 Background

Approaches to solving NP-hard problems include approximation algorithms with provable guarantees and heuristics tuned for empirical performance [20, 36, 38, 15]. A variety of heuristics are employed in practice, including greedy algorithms, local search, genetic algorithms, simulated annealing, particle swarm optimization, and others. By and large, the heuristics are based on extensive manual tuning and domain expertise.

Learning-based approaches have the potential to yield more effective empirical algorithms for NP-hard problems by learning from large datasets. The learning procedure can detect useful patterns and leverage regularities in real-world data that may escape human algorithm designers. He et al. [19] learned a node selection policy for branch-and-bound algorithms with imitation learning. Silver et al. [34, 35] used reinforcement learning to learn strategies for the game Go that achieved unprecedented results. Vinyals et al. [37] developed a new neural network architecture called a pointer network, and applied it to small-scale planar Travelling Salesman Problem (TSP) instances with up to 50 nodes. Bello et al. [6] used reinforcement learning to train pointer networks to generate solutions for synthetic planar TSP instances with up to 100 nodes, and also demonstrated their approach on synthetic random Knapsack problems with up to 200 elements.

Most recently, Dai et al. [10] used reinforcement learning to train a deep Q-network (DQN) to incrementally construct solutions to graph-based NP-hard problems, and showed that this approach outperforms prior learning-based techniques. Our work is related, but differs in several key respects. First, we do not use reinforcement learning, which is known as a particularly challenging optimization problem. Rather, we show that very strong performance and generalization can be achieved with supervised learning, which benefits from well-understood and reliable solvers. Second, we use a different predictive model, a graph convolutional network [12, 24]. Third, we design and train the network to synthesize a diverse set of solutions at once. This is key to our approach and enables rapid exploration of the solution space.

A technical note by Nowak et al. [30] describes an application of graph neural networks to the quadratic assignment problem. The authors report experiments on matching synthetic random 50-node graphs and generating solutions for 20-node random planar TSP instances. Unfortunately, the results did not surpass classic heuristics [9] or the results achieved by pointer networks [37].

## 3 Preliminaries

NP-complete problems are closely related to each other and all can be reduced to each other in polynomial time. (Of course, not all such reductions are efficient.) In this work we focus on four canonical NP-hard problems [22].

**Maximal Independent Set (MIS).** Given an undirected graph, find the largest subset of vertices in which no two are connected by an edge.

**Minimum Vertex Cover (MVC).** Given an undirected graph, find the smallest subset of vertices such that each edge in the graph is incident to at least one vertex in the selected set.

**Maximal Clique (MC).** Given an undirected graph, find the largest subset of vertices that form a clique.

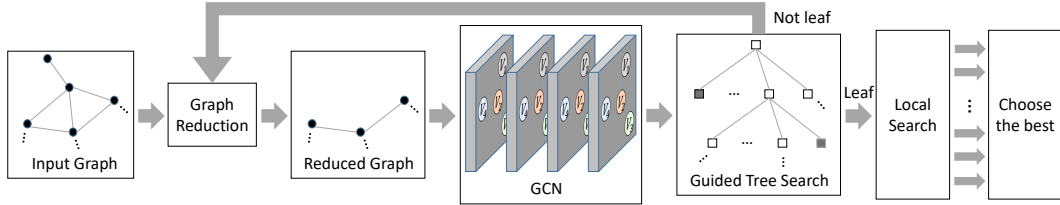

Figure 1: Algorithm overview. First, the input graph is reduced to an equivalent smaller graph. Then it is fed into the graph convolutional network $f$, which generates multiple probability maps that encode the likelihood of each vertex being in the optimal solution. The probability maps are used to iteratively label the vertices until all vertices are labelled. A complete labelling corresponds to a leaf in the search tree. Internal nodes in the search tree represent incomplete labellings that are generated along the way. The complete labellings generated by the tree search are refined by rapid local search. The best result is used as the final output.

**Satisfiability (SAT).** Consider a Boolean expression that is built from Boolean variables, parentheses, and the following operators: AND (conjunction), OR (disjunction), and NOT (negation). Here a Boolean expression is a conjunction of clauses, where a clause is a disjunction of literals. A literal is a Boolean variable or its negation. The problem is to find a Boolean labeling of all variables such that the given expression is true, or determine that no such label assignment exists.

All these problems can be reduced to each other. In particular, the MVC, MC, and SAT problems can all be represented as instances of the MIS problem, as reviewed in the supplementary material. Thus, Section 4 will focus primarily on the MIS problem, although the basic structure of the approach is more general. The experiments in Section 5 will be conducted on benchmarks and datasets for all four problems, which will be solved by converting them and solving the equivalent MIS problem.

## 4   Method

Consider a graph $\mathcal{G} = (\mathcal{V}, \mathcal{E}, \mathbf{A})$, where $\mathcal{V} = \{v_i\}_{i=1}^N$ is the set of $N$ vertices in $\mathcal{G}$, $\mathcal{E}$ is the set of $E$ edges, and $\mathbf{A} \in \{0, 1\}^{N \times N}$ is the corresponding unweighted symmetric adjacent matrix. Given $\mathcal{G}$, our goal is to produce a binary labelling for each vertex in $\mathcal{G}$, such that label 1 indicates that a vertex is in the independent set and label 0 indicates that it's not.

A natural approach to this problem is to train a deep network of some form to perform the labelling. That is, a network $f$ would take the graph $\mathcal{G}$ as input, and the output $f(\mathcal{G})$ would be a binary labelling of the nodes. A natural output representation is a probability map in $[0, 1]^N$ that indicates how likely each vertex is to belong to the MIS. This direct approach did not work well in our experiments. The problem is that converting the probability map $f(\mathcal{G})$ to a discrete assignment generally yields an invalid solution. (A set that is not independent.) Instead, we will use a network $f$ within a tree search procedure.

We begin in Section 4.1 by describing a basic network architecture for $f$. This network generates a probability map over the input graph. The network is used in a basic MIS solver that leverages it within a greedy procedure. Then, in Section 4.2 we modify the architecture and training of $f$ to synthesize multiple diverse probability maps, and leverage this within a more powerful tree search procedure. Finally, Section 4.3 describes two ideas adopted from classic heuristics that are complementary to the application of learning and are useful in accelerating computation and refining candidate solutions. The overall algorithm is illustrated in Figure 1.

### 4.1   Initial approach

We begin by describing a basic approach that introduces the overall network architecture and leads to a basic MIS solver. This will be extended into a more powerful solver in Section 4.2.

Let $\mathcal{D} = \{(\mathcal{G}_i, \mathbf{l}_i)\}$ be a training set, where $\mathcal{G}_i$ is a graph as defined above and $\mathbf{l}_i \in \{0, 1\}^{N \times 1}$ is one of the optimal solutions for the NP-hard graph problem. $\mathbf{l}_i$ is a binary map that specifies which vertices are included in the solution. The network $f(\mathcal{G}_i; \boldsymbol{\theta})$ is parameterized by $\boldsymbol{\theta}$ and is trained to predict $\mathbf{l}_i$ given $\mathcal{G}_i$.

We use a graph convolutional network (GCN) architecture [12, 24]. This architecture can perform dense prediction over a graph with pairwise edges. (See [7, 14] for overviews of related architectures.)

A GCN consists of multiple layers $\{\mathbf{H}^l\}$ where $\mathbf{H}^l \in \mathbb{R}^{N \times C^l}$ is the feature layer in the $l$-th layer and $C^l$ is the number of feature channels in the $l$-th layer. We initialize the input layer $\mathbf{H}^0$ with all ones and $\mathbf{H}^{l+1}$ is computed from the previous layer $\mathbf{H}^l$ with layer-wise convolutions:

$$\mathbf{H}^{l+1} = \sigma(\mathbf{H}^l\boldsymbol{\theta}_0^l + \mathbf{D}^{-\frac{1}{2}}\mathbf{A}\mathbf{D}^{-\frac{1}{2}}\mathbf{H}^l\boldsymbol{\theta}_1^l), \tag{1}$$

where $\boldsymbol{\theta}_0^l \in \mathbb{R}^{C^l \times C^{l+1}}$ and $\boldsymbol{\theta}_1^l \in \mathbb{R}^{C^l \times C^{l+1}}$ are trainable weights in the convolutions of the network, $\mathbf{D}$ is the degree matrix of $\mathbf{A}$ with its diagonal entry $\mathbf{D}(i,i) = \sum_j \mathbf{A}(j,i)$, and $\sigma(\cdot)$ is a nonlinear activation function (ReLU [29]). For the last layer $\mathbf{H}^L$, we do not use ReLU but apply a sigmoid to get a likelihood map.

During training, we minimize the binary cross-entropy loss for each training sample $(\mathcal{G}_i, \mathbf{l}_i)$:

$$\ell(\mathbf{l}_i, f(\mathcal{G}_i; \boldsymbol{\theta})) = \sum_{j=1}^{N}\{\mathbf{l}_{ij}\log(f_j(\mathcal{G}_i; \boldsymbol{\theta})) + (1 - \mathbf{l}_{ij})\log(1 - f_j(\mathcal{G}_i; \boldsymbol{\theta}))\}, \tag{2}$$

where $\mathbf{l}_{ij}$ is the $j$-th element of $\mathbf{l}_i$ and $f_j(\mathcal{G}_i; \boldsymbol{\theta})$ is the $j$-th element of $f(\mathcal{G}_i; \boldsymbol{\theta})$.

The output $f(\mathcal{G}_i; \boldsymbol{\theta})$ of a trained network is generally not a binary vector but real-valued vector in $[0,1]^N$. Simply rounding the real values to 0 or 1 may violate the independence constraints. A simple solution is to treat the prediction $f(\mathcal{G}_i; \boldsymbol{\theta})$ as a likelihood map over vertices and use the trained network within a greedy growing procedure that makes sure that the constraints are satisfied.

In this setup, $f(\mathcal{G}; \boldsymbol{\theta})$ is used as the heuristic function for a greedy search algorithm for MIS. Given $\mathcal{G}$, the algorithm labels a batch of vertices with 1 or 0 recursively. First, we sort all the vertices in descending order based on $f(\mathcal{G})$. Then we iterate over the sorted list in order and label each vertex as 1 and its neighbors as 0. This process stops when the next vertex in the sorted list is already labelled as 0. We remove all the labelled vertices and the incident edges from $\mathcal{G}$ and obtain a residual graph $\mathcal{G}'$. We use $\mathcal{G}'$ as input to $f$, obtain a new likelihood map, and repeat the process. The complete basic algorithm, referred to as BasicMIS, is specified in the supplementary material.

## 4.2 Diversity and tree search

One weakness of the approach presented so far is that the network can get confused when there are multiple optimal solutions for the same graph. For instance, Figure 2 shows two equivalent optimal solutions that induce completely different labellings. In other words, the solution space is multimodal and there are many different modes that may be encountered during training. Without further provisions, the network may learn to produce a labelling that "splits the difference" between the possible modes. In the setting of Figure 2 this would correspond to a probability assignment of 0.5 to each vertex, which is not a useful labelling.

To enable the network to differentiate between different modes, we extend the structure of $f$ to generate multiple probability maps. Given the input graph $\mathcal{G}$, the revised network $f$ generates $M$ probability maps: $\langle f^1(\mathcal{G}_i; \boldsymbol{\theta}), \ldots, f^M(\mathcal{G}_i; \boldsymbol{\theta})\rangle$. To train $f$ to generate diverse high-quality probability maps, we adopt the hindsight loss [18, 8, 28]:

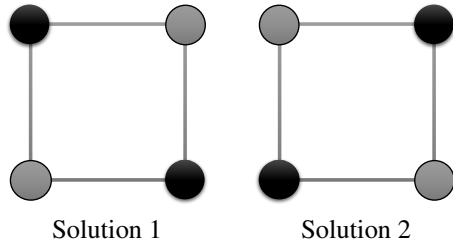

Solution 1          Solution 2

Figure 2: Two equivalent solutions for MIS on a four-vertex graph. The black vertices indicate the solution.

$$\mathcal{L}(\mathcal{D}, \boldsymbol{\theta}) = \sum_i \min_m \ell(\mathbf{l}_i, f^m(\mathcal{G}_i; \boldsymbol{\theta})), \tag{3}$$

where $\ell(\cdot, \cdot)$ is the binary cross-entropy loss defined in Equation 2. Note that the loss for a given training sample in Equation 3 is determined solely by the most accurate solution for that sample. This allows the network to spread its bets and generate multiple diverse solutions, each of which can be sharper.

Another advantage of producing multiple diverse probability maps is that we can explore multiple solutions with each run of $f$. Naively, we could apply the basic algorithm for each $f^m(\mathcal{G}_i; \boldsymbol{\theta})$, generating at least $M$ solutions. We can in principle generate exponentially many solutions, since in each iteration we can get $M$ probability maps for labelling the graph. We do not generate an exponential number of solutions, but leverage the new $f$ within a tree search procedure that generates a large number of solutions.

Ideally, we want to explore a large amount of diverse solutions in a limited time and choose the best one. The basic idea of the tree search algorithm is that we maintain a queue of incomplete solutions and randomly choose one of them to expand in each step. When we expand an incomplete solution, we use $M$ probability maps $\langle f^1(\mathcal{G}_i; \boldsymbol{\theta}), \dots, f^M(\mathcal{G}_i; \boldsymbol{\theta}) \rangle$ to spawn $M$ new more complete solutions, which are added to the queue. This is akin to breadth-first search, rather than depth-first search. If we expand the tree in depth-first fashion, the diversity of solutions will suffer as most of them have the same ancestors. By expanding the tree in breadth-first fashion, we can get higher diversity. To this end, the expanded tree nodes are kept in a queue and one is selected at random in each iteration for expansion. On a desktop machine used in our experiments, this procedure yields up to 20K diverse solutions in 10 minutes for a graph with 1,000 vertices. The revised algorithm is summarized in the supplement.

The presented tree search algorithm is inherently parallelizable, and can thus be significantly accelerated. The basic idea is to run multiple threads that choose different incomplete solutions from the queue and expand them. The parallelized tree search algorithm is summarized in the supplement. On the same desktop machine, the parallelized procedure yields up to 100K diverse solutions in 10 minutes for a graph with 1,000 vertices.

### 4.3 Classic elements

**Local search.** In the literature on approximation algorithms for NP-hard problems, there are useful heuristic strategies that modify a solution locally by simply inserting, deleting, and swapping nodes such that the solution quality can only improve [5, 16, 2]. We use this approach to refine the candidate solutions produced by tree search. Specifically, we use a 2-improvement local search algorithm [2, 13]. More details can be found in the supplement.

**Graph reduction.** There are also graph reduction techniques that can rapidly reduce a graph to a smaller one [1, 26] while preserving the size of the optimal MIS. This accelerates computation by only applying $f$ to the "complex" part of the graph. The reduction techniques we adopted are described in the supplement.

## 5 Experiments

### 5.1 Experimental setup

**Datasets.** For training, we use the SATLIB benchmark [21]. This dataset provides 40,000 synthetic 3-SAT instances that are all satisfiable; each instance consists of about 400 clauses with 3 literals. We convert these SAT instances to equivalent MIS graphs, which have about 1,200 vertices each. We will show that a network trained on these graphs generalizes to other problems, datasets, and to much larger graphs. We partition the dataset at random into a training set of size 38,000, a validation set of size 1,000, and a test set of size 1,000. The network trained on this training set will be applied to all other problems and datasets described below.

We evaluate on other problems and datasets as follows:

- SAT Competition 2017 [4]. The SAT Competition is a competitive event for SAT solvers. It was organized in conjunction with an annual conference on Theory and Applications of Satisfiability Testing. We evaluate on the 20 instances with the same scale as those in SATLIB. Note that small-scale does not necessarily mean easy. We evaluate SAT on this dataset in addition to the SATLIB test set.

- BUAA-MC [39]. This dataset includes 40 hard synthetic MC instances. These problems are specifically designed to be challenging [39]. The basic idea of generating hard instances is hiding the optimal solutions in random instances. We evaluate MC, MVC, and MIS on this dataset.

- SNAP Social Networks [27]. This dataset is part of the Stanford Large Network Dataset Collection. It includes real-world graphs from social networks such as Facebook, Twitter, Google Plus, etc. (Nodes are people, edges are interactions between people.) We use all social network graphs with less than a million nodes. The largest graph in the dataset we use has roughly 100,000 vertices and more than 10 million edges. We treat all edges as undirected. Details of the graphs can be found in the supplement. We evaluate MVC and MIS on this dataset.

- Citation networks [33]. This dataset includes real-world graphs from academic search engines. In these graphs, nodes are documents and edges are citations. We treat all edges as undirected. Details of the graphs can be found in the supplement. We evaluate MVC and MIS on this dataset.

**Baselines.** We mainly compare the presented approach to the recent deep learning method of Dai et al. [10]. This approach is referred to as S2V-DQN, following their terminology. For a number of

experiments, we will also show the results of this approach when it is enhanced by the graph reduction and local search procedures described in Section 4.3. This will be referred to as S2V-DQN+GR+LS. Following Dai et al. [10], we also list the performance of a classic greedy heuristic, referred to as Classic [31], and its enhanced version – Classic+GR+LS. In addition, we calibrate these results against three powerful alternative methods: a Satisfiability Modulo Theories (SMT) solver called Z3 [11], a SOTA MIS solver called ReduMIS [26], and a SOTA integer linear programming (ILP) solver called Gurobi [17].

**Network settings.** Our network has $L = 20$ graph convolutional layers, which is deep enough to get a large receptive field for each node in the input graph. Since our input is a graph without any feature vectors on vertices, the input $\mathbf{H}^0$ contains all-one vectors of size $C^0 = 32$. This input leads the network to treat all vertices equally, and thus the prediction is made based on the structure of the graph only. The widths of the intermediate layers are identical: $C^l = 32$ for $l = 1, \ldots, L - 1$. The width of the output layer is $C^L = M$, where $M$ is the number of output maps. We use $M = 32$. (Experiments indicate that performance saturates at $M = 32$.)

**Training.** Since SATLIB consists of synthetic SAT instances, the groud-truth assignments are known. With the ground-truth assignments, we can generate multiple labelling solutions for the corresponding graphs by switching on and off the free variables in a clause. We use Adam [23] with single-graph mini-batches and learning rate $10^{-4}$. Training proceeds for 200 epochs and takes about 16 hours on a desktop with an i7-5960X 3.0 GHz CPU and a Titan X GPU. S2V-DQN is trained on the same dataset with the same number of iterations.

**Testing.** For SAT, we report the number of problems that are solved by the evaluated approaches. This is a very important metric, because there is a big difference in applications between finding a satisfying assignment or not. It is a binary success/failure outcome. Since we solve the SAT problems via solving the equivalent MIS problems, we also report the size of the independent set that is found by the evaluated approaches. Note that it usually takes great effort to increase the size by 1 when the solution is close to the optimum, and thus small increases in the average size, on the order of 1, should be regarded as significant. For MVC, MIS, and MC, we report the size of the set identified by the evaluated approaches. On the BUAA-MC dataset, we also report the fraction of MC problems that are solved by the different approaches.

## 5.2 Results

We test all approaches on the same desktop with an i7-5960X 3.0 GHz CPU and a Titan X GPU. Our tree search algorithm is parallelized with 16 threads. Since the search will continue as long as allowed for Z3, Gurobi, ReduMIS, and our approach, we set a time limit. For fair comparison, we give the other methods $16\times$ running time, though we don't reboot them if they terminate earlier based on their stopping criteria. On the SATLIB and SAT Competition 2017 datasets, the time limit is 10 minutes. On the SNAP-SocialNetwork and CitationNetwork datasets with large graphs, the time limit is 30 minutes. There is no time limit for the Classic approach and S2V-DQN, since they only generate one solution. However, note that on SAT problems these approaches can terminate as soon as a satisfying assignment is found. Thus, on the SAT problems we report the median termination time.

| Method | Solved | MIS | Time (s) |
|---|---|---|---|
| Classic | 0.0% | 403.98 | 0.31 |
| Classic+GR+LS | 7.9% | 424.82 | 0.45 |
| S2V-DQN | 0.0% | 413.77 | 2.26 |
| S2V-DQN+GR+LS | 8.9% | 424.98 | 2.41 |
| Gurobi | 98.5% | 426.86 | 175.83 |
| Z3 | **100.0%** | – | **0.01** |
| ReduMIS | **100.0%** | **426.90** | 47.79 |
| Ours | **100.0%** | **426.90** | 11.47 |

Table 1: Results on the SATLIB test set. Fraction of solved SAT instances, average independent set size, and runtime.

| Method | Solved | MIS | Time (s) |
|---|---|---|---|
| Classic | 0.0% | 453.25 | 0.30 |
| Classic+GR+LS | 75.0% | 491.05 | 0.45 |
| S2V-DQN | 0.0% | 462.05 | 2.19 |
| S2V-DQN+GR+LS | 80.0% | 491.50 | 2.37 |
| Gurobi | 80.0% | – | 141.66 |
| Z3 | **100.0%** | – | **0.01** |
| ReduMIS | **100.0%** | **492.85** | 21.90 |
| Ours | **100.0%** | **492.85** | 12.20 |

Table 2: Results on the SAT Competition 2017. Fraction of solved SAT instances, average independent set size, and runtime.

We begin by reporting results on the SAT datasets. For each approach, Table 1 reports the percentage of solved SAT instances and the average independent set size on the test set of the SATLIB dataset. Note that there are 1,000 instances in the test set. The Classic approach cannot solve a single problem. S2V-DQN, though it has been trained on similar graphs in the training set, does not solve a single problem either, possibly because the reinforcement learning procedure did not discover fully satisfying solutions during training. Looking at the MIS sizes reveals that S2V-DQN discovers solutions that are close but struggles to get to the optimum. This observation is consistent with the results reported in the paper [10]. With refinement by the same classic elements we use, S2V-DQN+GR+LS solves 89 SAT instances out of 1,000, and Classic+GR+LS solves 79 SAT instances. In contrast, our approach solves all 1,000 SAT instances, which is slightly better than the SOTA ILP solver (Gurobi), and same as the modern SMT solver (Z3) and the SOTA MIS solver (ReduMIS). Note that Z3 directly solves the SAT problem and cannot solve any transformed MIS problem on the SATLIB dataset.

We also analyze the effect of the number $M$ of diverse solutions in our network. Note that this is analyzed on the single-threaded tree search algorithm, since the multi-threaded version solves all instances easily. Figure 3 plots the fraction of solved problems and average size of the computed MIS solution on the SATLIB validation set for $M = 1, 4, 32, 128, 256$. The results indicate that increasing the number of intermediate solutions helps up to $M = 32$, at which point the performance plateaus.

Table 2 reports results on SAT Competition 2017 instances. Again, both Classic and S2V-DQN solve 0 problems. When augmented by graph reduction and local search, Classic+GR+LS solve 75% of the problems, and S2V-DQN+GR+LS solves 80%, while our approach solves 100% of the problems. As sophisticated solvers, Z3 and ReduMIS solve 100%, while Gurobi solves 80%. Note that Gurobi cannot return a valid solution for some instances, and thus its independent set size is not listed.

Table 3 reports results on the BUAA-MC dataset. We evaluate MC, MIS, and MVC on this dataset. Since the optimal solutions for MC are given in this dataset, we report the fraction of MC problems solved optimally by each approach. Note that this dataset is designed to be highly challenging [39]. Most baselines, including Gurobi, cannot solve a single instance in this dataset. As a sophisticated MIS solver, ReduMIS solves 25%. S2V-DQN+GR+LS does not find any optimal solution on any problem instance. Our approach solves 62.5% of the instances. Note that our approach was only trained on synthetic SAT graphs from a different dataset. We see that the presented approach generalizes across datasets and problem types. We also evaluate MIS and MVC on these graphs. As shown in Table 3, our approach outperforms all the baselines on MIS and MVC.

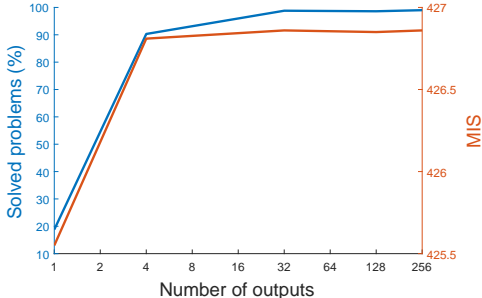

Figure 3: Effect of the hyperparameter $M$. The blue curve shows the fraction of solved problems on the SATLIB validation set for different settings of $M$. The orange curve shows the average size of the computed independent set for different settings of $M$.

| Method | Solved | MC | MIS | MVC |
|---|---|---|---|---|
| Classic | 0.0% | 30.03 | 21.53 | 991.72 |
| Classic+GR+LS | 0.0% | 42.83 | 24.64 | 988.61 |
| S2V-DQN | 0.0% | 40.40 | 23.76 | 989.49 |
| S2V-DQN+GR+LS | 0.0% | 42.98 | 24.70 | 988.55 |
| Gurobi | 0.0% | 39.75 | 24.12 | 989.13 |
| ReduMIS | 25.0% | 44.95 | 24.87 | 988.38 |
| Ours | **62.5%** | **45.55** | **25.06** | **988.19** |

Table 3: Results on the BUAA-MC dataset. The table reports the fraction of solved MC problems and the average size of MC, MIS, and MVS solutions.

| Method | Solved | MIS |
|---|---|---|
| Basic | 18.8% | 425.55 |
| Basic+Tree | 59.2% | 426.52 |
| No local search | 42.4% | 426.41 |
| No reduction | 91.0% | 426.81 |
| Full w/o parallel | 98.8% | 426.86 |
| Full with parallel | **100.0%** | **426.88** |

Table 4: Controlled experiment on the SATLIB validation set. The tables shows the fraction of solved SAT instances and the average independent set size.

| Name | MIS | | | | MVC | | | |
|---|---|---|---|---|---|---|---|---|
| | Classic | S2V-DQN | ReduMIS | Ours | Classic | S2V-DQN | ReduMIS | Ours |
| ego-Facebook | 993 | 1,020 | **1,046** | **1,046** | 3,046 | 3,019 | **2,993** | **2,993** |
| ego-Gplus | 56,866 | 56,603 | **57,394** | **57,394** | 50,748 | 51,011 | **50,220** | **50,220** |
| ego-Twitter | 36,235 | 36,275 | **36,843** | **36,843** | 45,071 | 45,031 | **44,463** | **44,463** |
| soc-Epinions1 | 53,457 | 53,089 | **53,599** | **53,599** | 22,422 | 22,790 | **22,280** | **22,280** |
| soc-Slashdot0811 | 53,009 | 52,719 | **53,314** | **53,314** | 24,351 | 24,641 | **24,046** | **24,046** |
| soc-Slashdot0922 | 56,087 | 55,506 | **56,398** | **56,398** | 26,081 | 26,662 | **25,770** | **25,770** |
| wiki-Vote | 4,730 | 4,779 | **4,866** | **4,866** | 2,385 | 2,336 | **2,249** | **2,249** |
| wiki-RfA | 8,019 | 7,956 | **8,131** | **8,131** | 2,816 | 2,879 | **2,704** | **2,704** |
| bitcoin-otc | 4,330 | 4,334 | **4,346** | **4,346** | 1,551 | 1,547 | **1,535** | **1,535** |
| bitcoin-alpha | 2,703 | 2,705 | **2,718** | **2,718** | 1,080 | 1,078 | **1,065** | **1,065** |

Table 5: Results on the SNAP Social Network graphs. The table lists the sizes of solutions for MIS and MVC found by the different approaches.

| Name | MIS | | | | MVC | | | |
|---|---|---|---|---|---|---|---|---|
| | Classic | S2V-DQN | ReduMIS | Ours | Classic | S2V-DQN | ReduMIS | Ours |
| Citeseer | 1,848 | 1,705 | **1,867** | **1,867** | 1,508 | 1,622 | **1,460** | **1,460** |
| Cora | 1,424 | 1,381 | **1,451** | **1,451** | 1,284 | 1,327 | **1,257** | **1,257** |
| Pubmed | 15,852 | 15,709 | **15,912** | **15,912** | 3,865 | 4,008 | **3,805** | **3,805** |

Table 6: Results on the citation networks.

Next we report results on large-scale real-world graphs. We use the different approaches to compute MIS and MVC on the SNAP Social Networks and the Citation Networks. The results are reported in Tables 5 and 6. Our approach and ReduMIS outperform the other baselines on all graphs. ReduMIS works as well as out approach, presumably because both methods find the optimal solutions on these graphs. Gurobi cannot return any valid solution for these large instances, and thus its results are not listed. One surprising observation is that S2V-DQN does not perform as well as the Classic approach when the graph size is larger than 10,000 vertices; the reason could be that S2V-DQN does not generalize well to large graphs. These results indicate that our approach generalizes well across problem types and datasets. In particular, it generalizes from synthetic graphs to real ones, from SAT graphs to real-world social networks, and from graphs with roughly 1,000 to graphs with roughly 100,000 nodes and more than 10 million edges. This may indicate that there are universal motifs that are present in graphs and occur across datasets and scales, and that the presented approach discovers these motifs.

Finally, we conduct a controlled experiment on the SATLIB validation set to analyze how each component contributes to the presented approach. Note that this is also analyzed on the single-threaded tree search algorithm, as the multi-threaded version solves all instances easily. The result is summarized in Table 4. First, we evaluate the initial approach presented in Section 4.1, augmented by reduction and local search (but no diversity); we refer to this approach as Basic. Then we evaluate a different version of Basic that generates multiple solutions and conducts tree search via random sampling, but does not utilize the diversity loss presented in Section 4.2; we refer to this version as Basic+Tree. (Basic+Tree is structurally similar to our full pipeline, but does not use the diversity loss.) Finally, we evaluate two ablated versions of our full pipeline, by removing the local search or the graph reduction. Our full approach with and without parallelization is listed for comparison. This experiment demonstrates that all components presented in this paper contribute to the results.

# 6 Conclusion

We have presented an approach to solving NP-hard problems with graph convolutional networks. Our approach trains a deep network to perform dense prediction over a graph. We showed that training the network to produce multiple solutions enables an effective exploration procedure. Our approach combines deep learning techniques with classic algorithmic ideas. The resulting algorithm

convincingly outperforms recent work. A particularly encouraging finding is that the approach generalizes across very different datasets and to problem instances that are larger by orders of magnitude than ones it was trained on.

We have focused on the maximal independent set (MIS) problem and on problems that can be easily mapped to it. This is not a universal solution. For example, we did not solve Maximal Clique on the large SNAP Social Networks and Citation networks, because the complementary graphs of these large networks are very dense, and all evaluated approaches either run out of memory or cannot return a result in reasonable time (24 hours). This highlights a limitation of only training a network for one task (MIS) and indicates the desirability of applying the presented approach directly to other problems such as Maximal Clique. The structure of the presented approach is quite general and can be leveraged to train networks that predict likelihood of Maximal Clique participation rather than likelihood of MIS participation, and likewise for other problems. We see the presented work as a step towards a new family of solvers for NP-hard problems that leverage both deep learning and classic heuristics. We will release code to support future progress along this direction.

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
