[Supplementary Material]

# Supplementary Material for "Combinatorial Optimization with Graph Convolutional Networks and Guided Tree Search"

**Zhuwen Li**
Intel Labs

**Qifeng Chen**
HKUST

**Vladlen Koltun**
Intel Labs

## A    Problem reductions

As mentioned in the main text, the MVC, MC, and SAT problems can all be represented as instances of the MIS problem:

**MVC → MIS.** Given a graph, the minimum vertex cover and the maximal independent set are complementary. A vertex set is independent if and only if its complement is a vertex cover, and thus the solutions of MIS and MVC are complementary to each other [6].

**MC → MIS.** The maximal clique of a graph is the maximal independent set of the complementary graph [6].

**SAT → MIS.** Given a SAT instance, we construct a graph as follows. Each literal in a clause is a vertex in the graph. Vertices in the same clause are adjacent to each other in the graph. An edge is spanned between two vertices in different clauses if they represent literals where one is the negation of the other. With this graph, if we can find an independent set with size equal to the number of clauses, the formula is satisfiable; otherwise, it is not. The independent set also specifies the truth assignment to the variables [3].

## B    Algorithms

The complete basic algorithm is summarized in Algorithm 1, the revised algorithm with diversity and tree search is summarized in Algorithm 2, and the variant with parallelized tree search is summarized in Algorithm 3.

## C    Classic elements

### C.1    Local search

For local search, we adopt a 2-improvement local search algorithm [2, 4]. This algorithm iterates over all vertices in the graph and attempts to replace a 1-labelled vertex $v_i$ with two 1-labelled vertices $v_j$ and $v_k$. In MIS, $v_j$ and $v_k$ must be neighbors of $v_i$ that are 1-tight and not adjacent to each other. Here, a vertex is 1-tight if exactly one of its neighbors is labelled 1. In other words, $v_i$ is the only 1-labelled neighbor of $v_j$ and $v_k$ in the graph. By using a data structure that allows inserting and deleting nodes in time proportional to their degrees, this local search algorithm can find a valid 2-improvement in $O(E)$ time if it exits. An even faster incremental version of the algorithm maintains a list of candidate nodes that are involved in 2-improvements. It ensures that a node is not repeatedly examined unless there is some change in its neighborhood.

### C.2    Graph reduction

For graph reduction, some efficient reduction techniques [1, 5] we adopted are given below:

- Pendant vertices: A vertex $v$ of degree one is called a pendant, and it must be in some MIS. Thus, $v$ and its neighbors can be removed from $\mathcal{G}$.

- Vertex folding: For a 2-degree vertex $v$ whose neighbors $u$ and $w$ are not adjacent, either $v$ is in some MIS, or both $u$ and $w$ are in some MIS. Thus, we can merge $u$, $v$ and $w$ to a single vertex $v'$ and decide which vertices are in the MIS later.

**Algorithm 1** BasicMIS
***
**Input:** Graph $\mathcal{G}$
**Output:** Labelling of all the vertices in $\mathcal{G}$
  1:  $v \leftarrow \text{argsort}(f(\mathcal{G}; \boldsymbol{\theta}))$ in a descending order
  2: **for** $i := 1$ **to** $\|v\|$ **do**
  3:     **if** vertex $v(i)$ is already labelled in $\mathcal{G}$ **then**
  4:        break
  5:     **end if**
  6:     Label $v(i)$ as 1 in $\mathcal{G}$
  7:     Label the neighors of $v(i)$ as 0 in $\mathcal{G}$
  8: **end for**
  9: $\mathcal{G}' \leftarrow$ residual graph of $\mathcal{G}$ by removing labelled vertices
10: **if** $\mathcal{G}'$ is not empty **then**
11:     Run BasicMIS on $(\mathcal{G}')$
12: **end if**
***

**Algorithm 2** MIS with diversity and tree search
***
**Input:** Graph $\mathcal{G}$
**Output:** The best solution found
  1: Initialize queue $\mathcal{Q}$ with $\mathcal{G}$
  2: **while** execute time is under budget **do**
  3:     $\mathcal{G}' \leftarrow \mathcal{Q}.\text{random\_pop}()$
  4:     **for** $m := 1$ **to** $M$ **do**
  5:        $v \leftarrow \text{argsort}(f^m(\mathcal{G}'; \boldsymbol{\theta}))$ in a descending order
  6:        **for** $i := 1$ **to** $\|v\|$ **do**
  7:           **if** vertex $v(i)$ is already labelled in $\mathcal{G}'$ **then**
  8:              break
  9:           **end if**
10:           Label $v(i)$ as 1 in $\mathcal{G}'$
11:           Label the neighors of $v(i)$ as 0 in $\mathcal{G}'$
12:        **end for**
13:        **if** $\mathcal{G}'$ is completely labelled **then**
14:           Update the current best solution
15:        **else**
16:           Remove labelled vertices from $\mathcal{G}'$
17:           Add $\mathcal{G}'$ to $\mathcal{Q}$
18:        **end if**
19:     **end for**
20: **end while**
***

- Unconfined: Define $\mathcal{N}(\cdot)$ as the neighbours of a vertex or set. A vertex $v$ is unconfined when determined by the following rules. First, let set $\mathcal{S} = \{v\}$. Then, we find a $u \in \mathcal{N}(\mathcal{S})$ such that $|\mathcal{N}(u) \cap \mathcal{S}| = 1$ and $|\mathcal{N}(u) \setminus \mathcal{S}|$ is minimized. If there is no such vertex, then $v$ is confined. If $\mathcal{N}(u) \setminus \mathcal{S} = \Phi$, then $v$ is unconfined. If $\mathcal{N}(u) \setminus \mathcal{S}$ is a single vertex $w$, then add $w$ to $\mathcal{S}$ and repeat the algorithm. Since there always exists an MIS without no unconfined vertices, they can be removed from $\mathcal{G}$.

- Twin: Define $\mathcal{G}[\mathcal{S}]$ as a subgraph of $\mathcal{G}$ induced by a set of vertices $\mathcal{S}$. Let $u$ and $v$ be vertices of degree 3 with $\mathcal{N}(u) = \mathcal{N}(v)$. If $\mathcal{G}[\mathcal{N}(u)]$ has edges, $u$ and $v$ must be in the MIS. If $\mathcal{G}[\mathcal{N}(u)]$ has no edges, some vertices in $\mathcal{N}(u)$ may belong to some MIS. In this case, we can still remove $u$, $v$, $\mathcal{N}(u)$ and $\mathcal{N}(v)$ from $G$, and add a new gadget vertex $w$ to $\mathcal{G}$ with edges to $u$'s order-2 neighbors (vertices at a distance 2 from u). Later if $w$ is in the computed MIS, then none of $u$'s order-2 neighbors are in the MIS, and therefore $\mathcal{N}(u)$ is in the MIS. If $w$ is not in the computed MIS, then some of $u$'s order-2 neighbors are in the MIS, and therefore $u$ and $v$ are in the MIS.

## D   Real-world graphs

Table 1 provides the statistics and descriptions of the real-world graphs used in our experiments.

**Algorithm 3** MIS with diversity and parallelized tree search
___

**Input:** Graph $\mathcal{G}$, thread number $T$
**Output:** The best solution found
1: **while** execute time is under budget **do**
2:     **if** queue $\mathcal{Q}$ is empty **then**
3:         Initialize $\mathcal{Q}$ with $\mathcal{G}$
4:     **end if**
5:     **for** all thread $t := 1$ **to** $T$ **do**
6:         $\mathcal{G}' \leftarrow \mathcal{Q}.\text{random\_pop}()$
7:         $m := \text{random\_integer}(M)$
8:         $v \leftarrow \text{argsort}(f^m(\mathcal{G}'; \boldsymbol{\theta}))$ in a descending order
9:         **for** $i := 1$ **to** $\|v\|$ **do**
10:            **if** vertex $v(i)$ is already labelled in $\mathcal{G}'$ **then**
11:              break
12:            **end if**
13:            Label $v(i)$ as 1 in $\mathcal{G}'$
14:            Label the neighors of $v(i)$ as 0 in $\mathcal{G}'$
15:         **end for**
16:         **if** $\mathcal{G}'$ is completely labelled **then**
17:            Update the current best solution
18:         **else**
19:            Remove labelled vertices from $\mathcal{G}'$
20:            Add $\mathcal{G}'$ to $\mathcal{Q}$
21:         **end if**
22:     **end for**
23: **end while**

| Name | Nodes | Edges | Description |
|---|---|---|---|
| ego-Facebook | 4,039 | 88,234 | Social circles from Facebook (anonymized) |
| ego-Gplus | 107,614 | 13,673,453 | Social circles from Google+ |
| ego-Twitter | 81,306 | 1,768,149 | Social circles from Twitter |
| soc-Epinions1 | 75,879 | 508,837 | Who-trusts-whom network of Epinions.com |
| soc-Slashdot0811 | 77,360 | 905,468 | Slashdot social network from November 2008 |
| soc-Slashdot0922 | 82,168 | 948,464 | Slashdot social network from February 2009 |
| wiki-Vote | 7,115 | 103,689 | Wikipedia who-votes-on-whom network |
| wiki-RfA | 10,835 | 159,388 | Wikipedia Requests for Adminship (with text) |
| bitcoin-otc | 5,881 | 35,592 | Bitcoin OTC web of trust network |
| bitcoin-alpha | 3,783 | 24,186 | Bitcoin Alpha web of trust network |
| Citeseer | 3,327 | 4,732 | Citation network from Citeseer |
| Cora | 2,708 | 5,429 | Citation network from Cora |
| Pubmed | 19,717 | 44,338 | Citation network from PubMed |

Table 1: Real-world graph statistics and descriptions.

## E   Network width $C$

We analyze the effect of the width $C$ of the intermediate layers. Table 2 shows the fraction of solved problems and average size of the computed MIS solution on the SATLIB validation set for $C = 16, 32, 64, 128$. The performance decreases when $C = 64, 128$, probably because a complex model is more computationally expensive and thus there is less time for the search algorithm. $C = 32$ provides the best balance of performance and efficiency.

## References

[1] Takuya Akiba and Yoichi Iwata. Branch-and-reduce exponential/FPT algorithms in practice: A case study of vertex cover. In *ALENEX*, 2015.

|  | $C = 16$ | $C = 32$ | $C = 64$ | $C = 128$ |
|---|---|---|---|---|
| Solved | 97.0% | **98.8%** | 97.6% | 97.2% |
| MIS | 426.86 | **426.88** | 426.87 | 426.87 |

Table 2: Effect of the hyperparameter $C$.

[2] Diogo Vieira Andrade, Mauricio G. C. Resende, and Renato Fonseca F. Werneck. Fast local search for the maximum independent set problem. *J. Heuristics*, 18(4), 2012.

[3] Sanjoy Dasgupta, Christos H. Papadimitriou, and Umesh V. Vazirani. *Algorithms*. McGraw-Hill, 2008.

[4] Thomas A. Feo, Mauricio G. C. Resende, and Stuart H. Smith. A greedy randomized adaptive search procedure for maximum independent set. *Operations Research*, 42(5), 1994.

[5] Sebastian Lamm, Peter Sanders, Christian Schulz, Darren Strash, and Renato F. Werneck. Finding near-optimal independent sets at scale. *J. Heuristics*, 23(4), 2017.

[6] Steven Skiena. *The Algorithm Design Manual*. Springer, 2008.