[Reviews · NeurIPS 2018]

Reviewer 1



This paper proposes a combinatorial optimization approach, wherein GCN is applied to compute the probability of each vertex being part of the optimal solution. The technicality focuses on the MIS problem and other problems are first equivalently reduced to MIS. Then, GCN is trained to output a probability map, where the supervision is the ground truth from a synthetic training set. If the problem has multiple optimal solutions, multiple probability maps are obtained. Finally, the maps are used to guide the tree search of optimal solution for new problem instances. The up side of this approach is that with a single synthetic training set, the trained network generalizes to other combinatorial problems, data sets, and much larger graphs. The experimental results are quite satisfactory. The down side of this approach is that it solely focuses on MIS and solves other problems through reduction. Sometimes the reduction may result in very dense graphs that pose memory/time bottleneck for graph neural networks. Overall, the paper demonstrates a successful use of GCN for combinatorial optimization. It also sets a good example for solving more than one NP-hard problem in a unified framework (through equivalence reduction). This work has good merits.

Reviewer 2



This paper proposes a method combining graph neural networks and guided tree search to tackle combinatorial optimization problems. The authors focus on the Maximum Independent Set (MIS), the Minimum Vertex Cover (MVC), Maximal Clique (MC) and Satisfiability which are all reduced to MIS. Given a MIS instance represented by a graph, the method consists in using the predictions of a graph convolutional neural (GCN) network, which is trained with binary cross-entropy to predict whether a vertex is in the maximum independent set, as input for a greedy search algorithm. The greedy search algorithm considers vertices in descending order of the GCN outputs, adding them to the independent set if doing so doesn't violate the independence assumption. Once this isn't possible, the process is repeated on the residual graph until termination. The method is further augmented with diversity, allowing the GCN to output different probability maps, and a tree search (partial solutions are added to a queue from which they will be randomly sampled to be further constructed). They train on a MIS graphs obtained from satisfiable SAT instances from the SATLIB benchmark for which solutions are available. They test the approach on instances coming from synthetic datasets and real-world graphs (such as citation networks or social networks). The authors compare their approach to the S2V-DQN baseline (from Learning Combinatorial Algorithms over Graph), the SOTA ILP solver Gurobi and the SMT solver Z3. Interestingly, the approach transfers well to different data distributions, larger instances and other problems. The proposed approach outperforms S2V-DQN and sometimes outperforms Z3 or Gurobi (which are hard baselines - assuming they're applied on the right problem). The paper is clearly written and proposes a novel approach for tackling combinatorial optimization (a fundamental field in computer science) using machine learning. Some results are particularly interesting such as the fact that the approach transfers that well to other problems and larger instances or the fact that using the diversity loss provides such an improvement. However, I have several issues with the current state of this draft (in no particular order). I am of course willing to change my low rating if these issues are addressed. - Although the proposed method clearly improves over Classic+GR+LS and in spite of the ablation study in Table 4, I think that a random baseline (ie Random+GR+LS) is necessary to evaluate the usefulness of the neural netwrok in the approach. That random baseline should be given the same time budget and should be similar to the proposed approach except for the fact that it wouldn't query the network (ie use a random order). - Supervised learning is presented as an advantage of the method on the premise that Reinforcement learning is a particularly challenging optimization problem and that supervised learning benefits from well-understood and reliable solvers. I believe that relying on labeled solutions is a weakness (albeit small) of the approach since it requires having solved (even approximately) NP-Hard problems. This is echoed in Neural Combinatorial Optimization with Reinforcement Learning (NCO), where the authors show that RL training clearly outperforms Supervised Learning training without requiring the need for costly solutions. - The S2V-DQN baseline is quite weak in my opinion: *) I take exception that S2V-DQN outperforms prior deep learning approaches such as NCO: the results that they report when reimplementing NCO are worse than those initially reported in NCO paper, they used a moving average baseline instead of a critic, they don't use the fact that NCO can sample many solutions as one etc... *) As correctly pointed out by the authors, S2V-DQN only outputs one solution while their approach considers multiple solutions. In comparison, NCO and Attention solves your TSP, approximately (which uses a Transformer-Pointer Network instead of an RNN Pointer Network) can output many solutions (when sampling) making them better baselines to the proposed approach. *) Attention solves your TSP, approximately does outperform prior learning-based approaches (and actually shows that the NCO approach outperforms S2V-DQN) This contrasts to the overall tone of the paper that their method clearly surpasses recent work ("Our algorithm convincingly outperforms recent work" in the conclusion) - The authors should also cite recent machine learning / combinatorial optimization work such as NeuroSAT, Attention solves your TSP, Learning the Multiple Traveling Salesmen Problem with Permutation Invariant Pooling Networks, etc. ================================================= ) Random+GR+LS: The 12.7% performance of this baseline outlines how important it is to consider multiple solutions. 2) Relying on supervised data is a weakness: I misunderstood that the labels relied on solvers. I agree that this is not an issue if the labels are generated without a solver and the authors have shown strong generalization. 3) The S2V-DQN baseline is quite weak. Out of the considered learning based approaches (ie NCO, ASTSP, the authors' method and S2V-DQN), S2V-DQN is the only method that considers a single solution. NCO and ASTSP can easily consider multiple solutions (by sampling) but it looks like the authors use the greedy version ASTSP only (this is hard to tell from the rebuttal but given that ASTSP solves 0% while random search solves 12.7% this seems to be the case). 4) I also invite the authors to reconsider the following statements which are incorrect / misleading. (a) "Most recently, Dai et al. [10] used reinforcement learning to train a deep Q-network (DQN) to incrementally construct solutions to graph-based NP-hard problems, and showed that this approach outperforms prior learning-based techniques." - For tasks where the methods are comparable (and compared), Dai et al report worse results when reimplementing NCO than those initially reported in NCO paper. Careful reading indicates that they used a moving average baseline instead of a critic, they don't use the fact that NCO can sample many solutions as one, etc... This is also echoed in a figure from the independent ASTSP paper which present NCO and ASTSP as outperforming Dai et al. (b) "First, we do not use reinforcement learning, which is known as a particularly challenging optimization problem. Rather, we show that very strong performance and generalization can be achieved with supervised learning, which benefits from well-understood and reliable solvers." - Previous work in combinatorial optimization has shown that RL is better suited than supervised learning (at least for pointer network models). - Reliance on solvers is a weakness in general. (c) (rebuttal) "RL-based approaches consider problems up to 100 variables". This is not true (although the authors have indeed tackled larger problems that prior work). It also not clear why this statement is relevant to the comparison of RL vs supervised learning: I think it has more to do with the types of models being used rather than the training method. In summary, I have a significant presentation issue with this paper. I agree that % of solved instances is an important practical metric but it obviously greatly favors methods that consider multiple solutions. The paper does a poor job at outlining this fact and compares only to learning based methods that only consider a single solution (ie S2V-DQN). The fact that all other learning based are outperformed by the random search further demonstrates my point. Methods tend to be much closer when looking at the average independent set size metric. (The authors also didn't report this metric for ASTSP or Random+GR+LS in the rebuttal). Relying on search (ie trying multiple solutions) to get good results is not a weakness of the paper in itself and I actually think it's a promising direction to combine both classic search methods with machine learning. However comparing to learning-based methods that only consider a single solution when considering to a metric that favors multiple solutions can be misleading . Especially that these methods can easily be augmented by search (ie sampling in NCO and ASTSP). The other incorrect statements about related work (mistakenly saying that the RL-based approaches consider up to 100 variables, etc.) also don't play in favor of the work. If this wasn't for these issues, this would be a strong paper. I appreciate the thorough response of the authors and I have updated my score to 4.

Reviewer 3



Post-rebuttal: Thank you for taking the time to run additional experiments for the rebuttal and answer our questions. Your answers to my questions are generally convincing. Regarding WalkSAT, the 100%-solved result goes to show that problem-specific heuristics can be very powerful. The same could be done for MIS, see [z] for example (with code). I hope that future versions of this paper include better baseline heuristics for comparison, and wish you luck. [z] Lijun Chang, Wei Li, Wenjie Zhang, "Computing A Near-Maximum Independent Set in Linear Time by Reducing-Peeling" Proceedings of the ACM SIGMOD International Conference on Management of Data (SIGMOD’17), 2017 Code for [z]: https://github.com/LijunChang/Near-Maximum-Independent-Set ------------------------------------- This paper proposes a new deep learning framework for generating solutions to some combinatorial optimization problems on graphs. The proposed method is supervised (instance-OptSolution pairs are provided in the training set), and uses graph convolutional networks (GCNs) as a deep learning model that maps an input graph to a latent space and then to a probability distribution over vertices. The distribution can then be used to generate a potential solution to the optimization problem. Since an instance may have multiple optima, learning to generate a single optimum may not generalize well. To address this issue, the first primary contribution of this paper is to generate multiple probability distributions, rather than a single one, and use the so-called "hindsight loss" for training. This allows the deep learning model to generate multiple guesses of what it thinks an optimum should look like, but the hindsight loss penalizes only for the best such guess produced. Another contribution is to hybridize the deep learning model with certain classical steps typically used in combinatorial optimization, namely graph reduction, tree search, and local search. In practice, the proposed method is used to solve the Maximum Independent Set (MIS) problem, and other graph problems are solved by reducing them to MIS, then transforming the MIS solution back to the original problem of interest. Experimentally, it is shown that the proposed method outperforms S2V-DQN, a reinforcement learning + graph embedding approach proposed recently. The proposed method also generalizes well across different sets of instances, as well as from small training instance to much larger test instances. An ablation and hyperparameter study illustrate that the various components of the proposed method all substantially contribute to its performance. Overall, I am positive about this submission and so my score is 7/10. I think the paper is well-written, features some methodological contributions, and shows promising experimental results. I do have some questions that I would like the authors to address. 1- Efficacy beyond MIS: First, as the authors note in their conclusion, a reduction to MIS may involve a quadratic number of variables in the original problem size, limiting the applicability of an MIS-trained GCN to other problems. Second, it should be noted that the proposed method works for subset-selection type problems, like MIS, but not for permutation-type problems like the TSP; the authors do not claim otherwise, but it is important to acknowledge this point or discuss straightforward extensions to the method that would make it work for say the TSP and similar problems. Last but not least, I am concerned that the proposed method may be really good for the MIS, but not for other problems. It is clear that a good MIS algorithm will result in good solutions to the original SAT instance, since a higher MIS objective makes it more likely that the SAT instance becomes satisfiable; the same can be said for MVC and MC, and the authors demonstrate great results in this setting. However, what if you were to apply your procedure to a problem other than MIS during training, as done in [6] (tsp and knapsack) or [10) (tsp, mvc, maxcut)? Answering this question is necessary to understanding the generality of the proposed method. 2- Comparison against SAT methods: for the SAT experiments, it only makes sense to compare against SAT algorithms, rather than MIS algorithms on the transformed instance. For example, one can use GSAT or WalkSAT heuristics, or their more advanced versions. Additionally, the authors should compare against NeuroSAT [2], which uses neural graph message-passing on the bipartite graph encoding of a SAT instance to classify it as sat/unsat and eventually produce a potential assignment. The same can be said about the MC/MVC experiment of tables 3, 5, 6. 3- Role of graph reduction: related to point 1 above, the graph reduction procedure is very specific to MIS, and is a performance booster based on Table 4. If one were to apply your method during training time to a problem other than MIS, would they be able to leverage similar graph reduction techniques? Another point here is related to your training set: the graph reduction may work better because of the structure of the MIS graph, since it encodes a SAT instance, which is a very structured object. I am not sure the graph reduction procedure would be as effective on other types of graphs. 4- Hyperparameter tuning: How do you select L=20, C^0=32? The choice of M=32 is well-justified by Figure 3, but the other two seem to be based on intuition. Please comment on any hyperparameter search you may have performed, or why you think such values are inherently good and will generalize to another training distribution. 5- Hindsight loss minimization: This is just my lack of expertise with this particular loss, but how do you optimize it given that the min operator makes the function non-differentiable? 6- Tree search or beam search? Tree search is usually associated with complete methods that are guaranteed to find an optimum and certify its optimality. What you use is more like beam search that decodes the M probability distributions; worth thinking about the right name here. 7- Missing references: [1] hybridizes a learned TSP policy with local search methods - worth to connect it with your hybridization approach. [3-6], similarly to reference [19] by He et al., use supervised learning for branching/primal heuristics in tree search. [1] Deudon, Michel, et al. "Learning Heuristics for the TSP by Policy Gradient." International Conference on the Integration of Constraint Programming, Artificial Intelligence, and Operations Research. Springer, Cham, 2018. [2] Selsam, Daniel, et al. "Learning a SAT Solver from Single-Bit Supervision." arXiv preprint arXiv:1802.03685 (2018). [3] Khalil, Elias Boutros, et al. "Learning to Branch in Mixed Integer Programming." AAAI. 2016. [4] Khalil, Elias B., et al. "Learning to run heuristics in tree search." 26th International Joint Conference on Artificial Intelligence (IJCAI). 2017. [5] Alvarez, Alejandro Marcos, Quentin Louveaux, and Louis Wehenkel. "A machine learning-based approximation of strong branching." INFORMS Journal on Computing 29.1 (2017): 185-195. Minor typos: 179: breath-first -> breadth-first 181: bread-first -> breadth-first